# Developing and validating a clinical prediction model to predict epilepsy-related emergency department attendance, hospital admission, or death: A cohort study protocol

Gashirai K. Mbizvo[1,2,3,4*], Glen P. Martin[5], Laura J. Bonnett[6], Pieta Schofield[7], Hilary Garret[8], Alan Griffiths[8], W Owen Pickrell[9,10], Iain Buchan[7‡], Gregory Y.H. Lip[1,11‡], Anthony G. Marson[2,3,4‡]

**1** Liverpool Centre for Cardiovascular Science at University of Liverpool, Liverpool John Moores University and Liverpool Heart & Chest Hospital, Liverpool, United Kingdom, **2** Liverpool Interdisciplinary Neuroscience Centre, University of Liverpool, Liverpool, United Kingdom, **3** Pharmacology and Therapeutics, Institute of Systems, Molecular and Integrative Biology, University of Liverpool, Liverpool, United Kingdom, **4** The Walton Centre NHS Foundation Trust, Liverpool, United Kingdom, **5** Division of Informatics, Imaging and Data Science, Faculty of Biology, Medicine and Health, University of Manchester, Manchester Academic Health Science Centre, Manchester, United Kingdom, **6** University of Liverpool Department of Biostatistics, Liverpool, United Kingdom, **7** Department of Public Health, Policy and Systems, Institute of Population Health, University of Liverpool, Liverpool, United Kingdom, **8** Public Involvement Team, Applied Research Collaboration North West Coast (ARC NWC), Liverpool, United Kingdom, **9** Neurology Research Group, Swansea University Medical School, Faculty of Medicine, Health and Life Science, Swansea University, Swansea, Wales, United Kingdom, **10** Morriston Hospital, Swansea Bay University Health Board, Swansea, Wales, United Kingdom, **11** Danish Centre for Health Services Research, Department of Clinical Medicine, Aalborg University, Aalborg, Denmark

‡ Joint senior authors
* Gashirai.Mbizvo@liverpool.ac.uk

## Abstract

### Introduction

This retrospective open cohort study develops and externally validates a clinical prediction model (CPM) to predict the joint risk of two important outcomes occurring within the next year in people with epilepsy (PWE). These are: A) seizure-related emergency department or hospital admission; and B) epilepsy-related death. This will provide clinicians with a tool to predict either or both of these common outcomes. This has not previously been done despite both being potentially avoidable, interrelated, and devastating for patients and their families. We hypothesise that the CPM will identify individuals at high or low risk of either or both outcomes. We will guide clinicians on proposed actions to take based on the overall risk score.

### Methods and analysis

Routinely collected, anonymised, electronic health data from the following research platforms will be used: i) Clinical Practice Research Datalink (CPRD); ii) Secure Anonymised Information Linkage databank (SAIL); iii) Combined Intelligence for Population

**Data availability statement:** No datasets were generated or analysed during the current study. All relevant data from this study will be made available upon study completion.

**Funding:** The author(s) received no specific funding for this work.

**Competing interests:** The authors have declared that no competing interests exist.

Health Action (CIPHA); and iv) TriNetX. We will study PWE aged ≥16 years having outcomes A and/or B between 2010–2024 within these datasets. Sample sizes of over 100,000 PWE are expected across these datasets. Candidate predictors will include demographic, lifestyle, clinical, and management variables. Logistic regression and multistate modelling will be used to develop a suitable CPM. The choice of modelling approach will be informed by consultation with clinicians and members of the public. We will assess the model's predictive performance using CPRD as a development dataset, and conduct external validation using SAIL, CIPHA, and TriNetX.

## Conclusions

This large study will develop and validate a CPM for PWE, creating an internationally generalisable tool for subsequent clinical implementation. It will predict the joint risk of acute admission and death in PWE. Mortality prediction is highlighted by NICE as a key recommendation for epilepsy research. The study has been co-developed by epilepsy researchers and members of the public affected by epilepsy.

## Introduction

Hospital admission and deaths from seizures are important outcomes for people with epilepsy (PWE). To inform patient counselling and clinical guidance about such outcomes, statistically robust and clinically meaningful clinical prediction models (CPMs) of who is at high (and low) risk of these outcomes can be developed and validated.

### Seizure-related emergency department or hospital admission

As we previously demonstrated [1], PWE are at substantially increased risk of seizure-related emergency department (ED) or hospital admission [2]. Indeed, seizures are the most common neurological cause of ED or hospital admission in England [2], and such admissions can be predictors of subsequent epilepsy-related death [3,4]. However, undergoing ED or hospital admission for epilepsy is often clinically unnecessary and typically leads to little benefit for management of the epilepsy because treatment decisions in epilepsy are complex and require specialist expertise, training and guidance [2]. The admitted PWE are usually seen by junior doctors and physicians without particular expertise in epilepsy and frequently discharged without specialist consultation or referral [1,2,5]. This represents a missed opportunity for risk mitigation at a time when this may be crucial [1]. Such admissions could be avoided through early prediction of high-risk groups, particularly if non-specialists such as general practitioners (GPs), paramedics, and physicians are equipped with tools to support early prediction. This is because epilepsy is an ambulatory-care-sensitive condition [2,6,7]; meaning that people at high-risk can often benefit from proactive management strategies. These include scheduled specialist and community follow-up, diversion to alternative care pathways, optimised self-management, and accelerated medication reviews [8,9].

## Epilepsy-related deaths

As we previously demonstrated [5,10], PWE are at significantly increased risk of premature death. Some of those deaths may be entirely unrelated to their epilepsy. However, a substantial proportion are epilepsy-related [10,11]. These are operationally defined as any death listing epilepsy as an underlying or contributory cause within death records [5,12,13], in line with national guidance [14]. Although sudden unexpected death in epilepsy (SUDEP) is a common cause of epilepsy-related death, it is not the only one. Our previous work has shown that other mechanisms are equally common. These include aspiration pneumonia, cardiac arrest, antiseizure medication (ASM) poisoning, drowning, and alcohol dependence [5,10]. Most epilepsy-related deaths occur in young adults [5,10]. When we looked at epilepsy-related deaths as a group, nearly 80% were potentially avoidable [5]. Therefore, it is becoming increasingly pragmatic to investigate epilepsy-related deaths as a group rather than focusing on each cause individually [11,12,15].

## Clinical prediction models (CPMs)

CPMs estimate the risk of a future clinical outcome in an individual patient based on information retrieved from their routinely collected health data. This might include information on age, sex, ethnicity, comorbidities, previous treatments, and other characteristics [16]. Therefore, CPMs help guide and standardise shared clinical decision-making and service delivery by drawing attention to high-risk individuals (needing earlier or more invasive treatments), and low-risk individuals (needing less invasive treatments with fewer side-effects) [17]. This helps personalise clinical management strategies. CPMs also help patients understand their own risks by quantifying them in a succinctly presented tool to aid patient-doctor discussions about risk [18]. For example, our $CHA_2DS_2$-VASc score [19,20] is a simple 7-item CPM using everyday clinical information to accurately predict risk of stroke after developing atrial fibrillation (AF). This helps guide subsequent anticoagulation treatment decisions and patient discussions. We made $CHA_2DS_2$-VASc freely available online to any clinician in the world at www.mdcalc.com. This combination of simplicity of use, accuracy, and wide availability has translated into substantial global impact. For example, use of this CPM is recommended within National Institute for Health and Care Excellence (NICE) guidelines [21].

## CPMs for clinically meaningful outcomes for PWE

An effective way to help avoid epilepsy-related deaths is to develop CPMs to identify high-risk groups and thereby aid clinicians in prioritising their care [4]. For example, in non-specialist settings, where such patients often first present [1,2], high CPM scores could prompt several important actions. These include revisiting discussions about seizure safety with patients and directing them to trusted online resources for support (e.g., www.epilepsy.org.uk/living/safety). The patients could also be referred to rapid-access epilepsy clinics, as opposed to reserving such clinics for first-fit patients alone. The potential impact of such approaches is well recognised. For example, an asthma death review highlighted that many deaths were potentially preventable with more proactive clinical care. This included ensuring prompt follow-up after hospital admission, and consistent implementation of established clinical guidelines [22]. In specialist settings, high CPM scores could prompt referral for epilepsy surgery earlier than would have otherwise been considered. They could also prompt discussion in a multidisciplinary team meeting, further medication reviews, and implementation of seizure alarms or nocturnal supervision.

There are currently no CPMs for ED or hospital admission in PWE. Work is planned to develop a risk prediction tool to estimate the benefits a person with epilepsy would receive if conveyed to ED and risks if not [8]. Additionally, there are few CPMs of deaths in PWE. Our recently developed Scottish Epilepsy Deaths Study score was a helpful step forward in attempting to predict epilepsy-related deaths as a group [4]. However, because it relied on hand-searched medical records, the sample size was to 224 cases and controls. This constrained the statistical power of the model, resulting in wider confidence intervals, reduced precision, and a maximum of four predictors. The SUDEP-7 and SUDEP-3 inventories

focused on predicting SUDEP alone [23], and therefore missed the remaining epilepsy-related deaths as a group. They were also drawn from small sample sizes of 19 and 28 patients, respectively, limiting their reliability. Similarly, the Personalised Prediction Tool [24] focused on SUDEP alone and had only 287 cases and 986 controls.

None of the above CPMs have been validated in independent, geographically distinct, datasets (so-called external validation), meaning they cannot currently be used in clinical practice [16]. The SUDEP and Seizure Safety Checklist [25] is expert consensus guidance rather than a CPM. Developing a validated CPM for epilepsy-related deaths was highlighted as a key recommendation for epilepsy research in recent NICE guidelines [26].

### Study aims and potential benefits to PWE

In line with the recommendations of NICE [26], we aim to develop and externally validate a CPM to predict the joint risk of two outcomes occurring within the next year in PWE [27]: A) seizure-related ED or hospital admission; and B) epilepsy-related death. We hypothesise the CPM will be able to identify individuals at high or low risk of either or both outcomes. This will give clinicians seeing PWE a validated tool to predict either or both of these important outcomes. This has not been done before despite both outcomes being common, potentially avoidable, devastating for patients and their families, and closely interrelated [1–5,9]. We will develop guidance for clinicians on proposed actions to take based on the overall risk score. When implemented, we expect this work to contribute to a reduction in the number of seizure-related ED or hospital admissions and epilepsy-related deaths.

## Methods

### Study design

This protocol has been developed in accordance with the Transparent Reporting of a multivariable prediction model for Individual Prognosis Or Diagnosis (TRIPOD) guidelines [28]. We will undertake a retrospective open cohort study [7] of anonymised health data from a range of electronic sources. These will include clinical, administrative, and socio-demographic records from general practices and healthcare organisations (HCOs). Specifically, we will use data from the research platforms listed below. Each contains data that are demographically representative of their respective general populations [3,13,29–31]:

- **Clinical Practice Resarch Datalink (CPRD)** in England, covering a population of approximately 60 million people [3,29];

- **Secure Anonymised Information Linkage (SAIL)** databank in Wales, covering 3.1 million people [13]

- **Combined Intelligence for Population Health Action (CIPHA)** research platform in Cheshire and Merseyside, covering 2.6 million people, accessed via www.cipha.nhs.uk. CIPHA a Trusted Research Environment that supports real-time population health analytics and was originally developed in response to the COVID-19 pandemic (Fig 1);

- **TriNetX** research platform [30]. TriNetX is an international research platform with data from ~250m patients from >215 HCOs across 19 countries predominantly in North America but also South America, Europe, the Middle East, Africa, and Asia Pacific [30,31]. It holds ~70 billion date- and patient-indexed clinical observations [30].

There are several reasons for planning to study CPRD, SAIL, CIPHA and TriNetX. First, it provides a contingency plan should we fail acquire one or more of the datasets, e.g., owing to cost. Second, using these geographically distinct datasets enables both internal and external validation. This enhances the generalisability of the final model, and supports its future implementation in diverse clinical settings. In particular, each dataset's coverage adds value to understanding the model's performance across different healthcare systems and demographics. For example, CPRD is strengthened by its extensive primary care representation, SAIL by its linkage to socio-demographic data, CIPHA allows real-time analytics

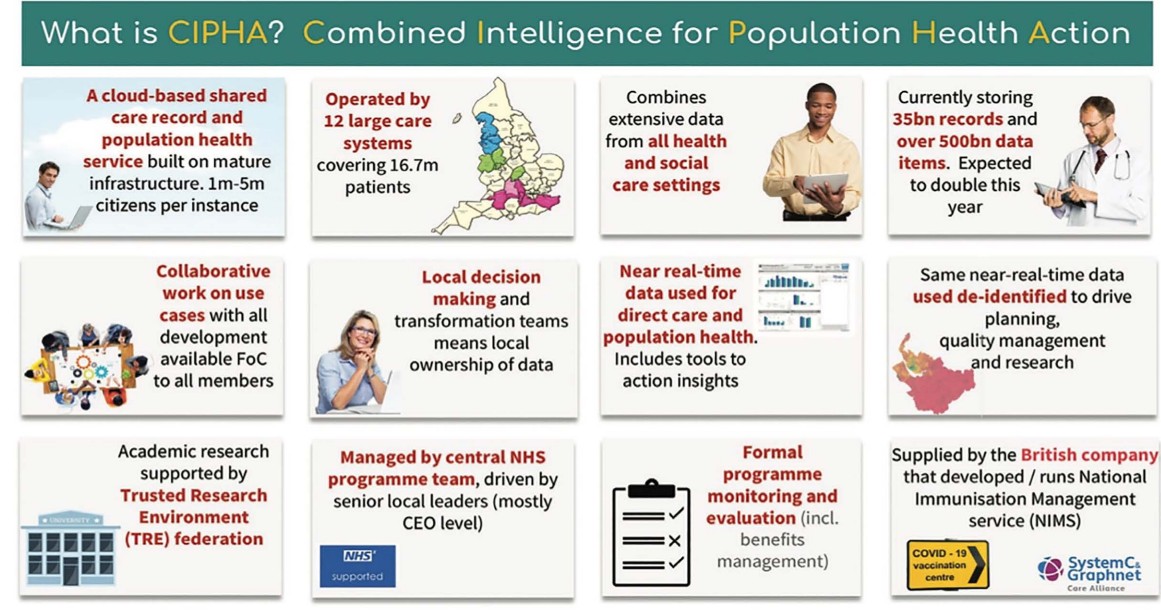

**Fig 1. CIPHA research platform summary (reproduced with permission) [32].**

with re-identification at source, and TriNetX has a substantial sample size, offering flexibility in selecting a large number of predictors.

Raw person-level data will be imported from each database following the relevant information governance training and data access protocol approval. Imported data will be linked from primary care, secondary care (ED, inpatients, and outpatients), deprivation, and mortality datasets. They will be analysed through 01/01/2010–31/10/2024 to study data from adults aged ≥16 years, capturing peak mortality risks [5].

Drawing lessons from the limitations highlighted in current CPM literature in the area [4,23,24], our study represents a paradigm shift by using large routinely-collected health research datasets, allowing our CPM to be developed using data from hundreds of thousands of PWE. Furthermore, we will undertake both internal and external validation of our CPM within the same project, thereby facilitating clinical implementation directly from our work [16,19,20]. We will prioritise variables readily available in primary care, hospital and outpatient settings to help maximise clinical utility both amongst specialists and non-specialists [7,19,20]. We will restrict size of the CPM to no more than 7 variables [19,20] to maximise ease of use for clinicians using the model iteratively [19,20] (Fig 2A) and avoid statistical overfitting [33]. We will also make the CPM globally accessible by depositing it online [19,20]. We will focus on including predictor variables that have been newly acquired in the preceding year so that the CPM can also act as a dynamic alert system for the acquisition of important new risks when patients are seen at, e.g., annual reviews. Furthermore, we will work with GPs and electronic health record (EHR) vendors such as EMIS® to ensure the CPM is designed in such a way as to facilitate future integration into electronic primary care systems, allowing pre-emptive risk stratification with automated alerts (Fig 2B), with clear action plans developed through GP, specialist, and public consultation.

## Case ascertainment

We will apply our previously validated [34,35] epilepsy symptom and disease codes combined with ASMs to identify PWE within CPRD (Aurum), SAIL, CIPHA, and TriNetX. These have positive predictive values and sensitivities >90% and >80%,

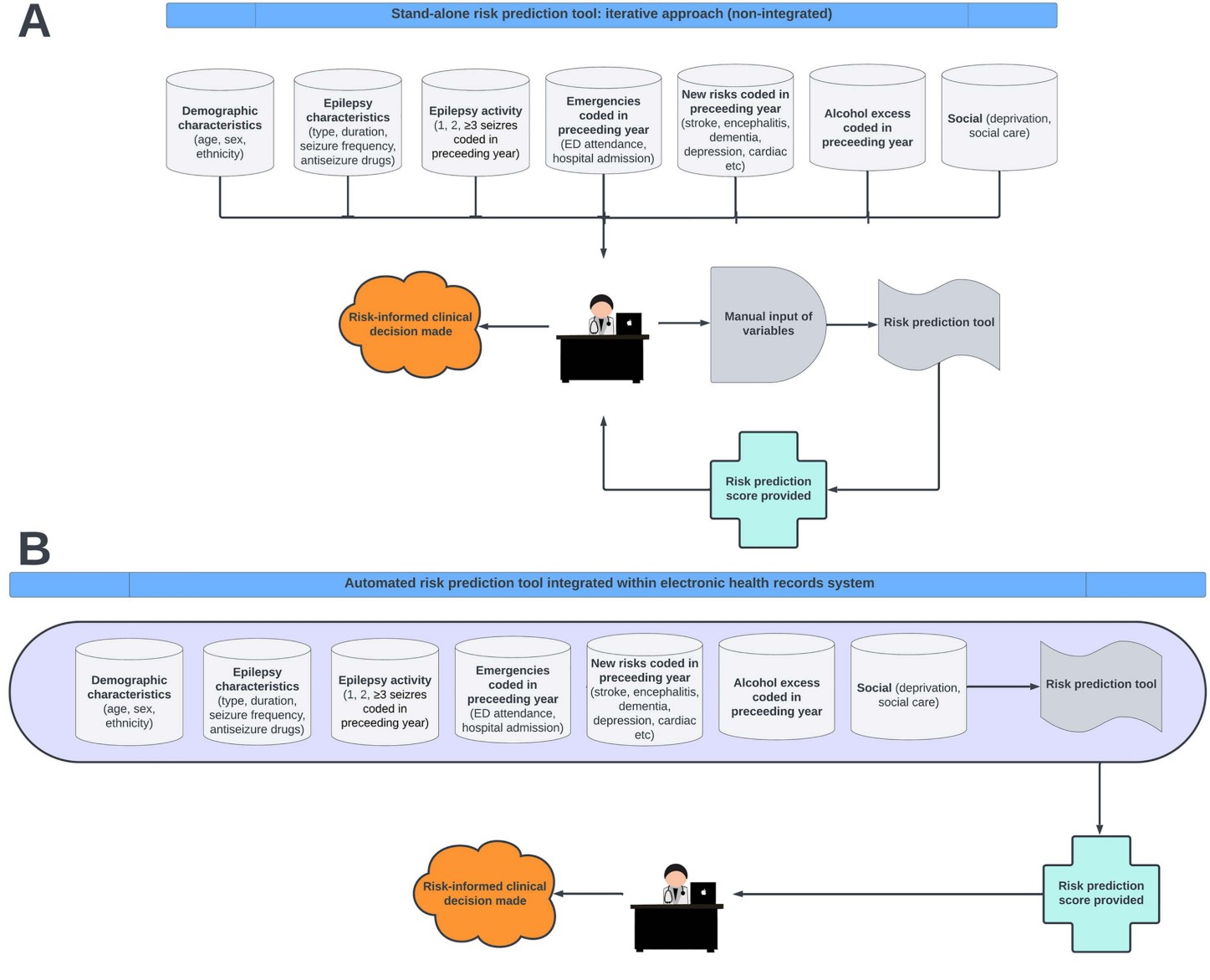

**Fig 2. Future clinical implementation options for the proposed clinical prediction model.** *Key:* A = iterative design, B = automated design.

respectively, for identifying epilepsy within administrative datasets [34,35]. Using these codes, our pre-submission feasibility search identified a large cohort of 227,271 PWE aged ≥16 years within CPRD between 01/01/2010–31/12/2021. This contains data from both incident and prevalent cases, which will be analysed as a single cohort and in respective subgroups.

## Outcomes of interest

We will define epilepsy-related death as where seizures or epilepsy are the underlying or contributory causes of death (International Classification of Diseases (ICD)-10 codes G40–41, R56.8) [5,13,34,35]. The same codes will be used to identify seizure-related ED or hospital admissions [34,35]. All codes will be combined with a requirement for co-prescribed ASMs to optimise diagnostic accuracy [13,34,35].

Two binary outcomes will be assessed between 01/01/2010–31/10/2024: A) first seizure-related ED or hospital admission; and B) epilepsy-related death. Data from PWE experiencing either or both or neither outcomes will be modelled.

**Predictor variables**

Candidate SNOMED-/Read-/ICD-coded predictors will be derived from existing literature [3,5,7,10,36] and informed by workshop-based consultation with clinicians and members of the public affected by epilepsy. They will include the following (with continuous variables analysed as such to maximise statistical power, but presented as categories post-hoc to maximise clinical interpretation):

- Age;

- Sex;

- Ethnicity (minority/non-minority);

- Deprivation quintile – Small area level/Welsh Index datasets;

- Cambridge Multimorbidity Score [37];

- Learning difficulties or developmental delay (yes/no);

- Alcohol excess coded in preceding year (yes/no);

- Type of epilepsy (focal-/generalised-/unknown-onset);

- Epilepsy duration (incident ≤1 year/prevalent >1 year);

- Management coded in preceding year:

  ◦ Seen by GP about epilepsy (0, 1, 2, ≥3 times);

  ◦ Seen in a neurology clinic (yes/no) – outpatient dataset;

  ◦ ASM number (1, 2, ≥3);

- Emergencies – ED and hospital admission datasets;

  ◦ Seizure-related ED or hospital admissions in year before study entry (0, 1, 2, ≥3);

  ◦ ED or hospital admissions unrelated to seizures in year preceding outcome index (0, 1, 2, ≥3);

- Neurological risks coded in preceding year:

  ◦ 0, 1, 2, ≥3 seizures;

  ◦ New stroke/depression/psychosis/dementia/autoimmune encephalitis;

- Other new risks coded in preceding year:

  ◦ *Cardiovascular:* Myocardial infarction/atrial fibrillation (AF)/cardiac failure;

  ◦ *Respiratory:* Chronic obstructive pulmonary disease (COPD);

  ◦ *Metabolic:* Diabetes/extreme body mass index (BMI: <18.5, ≥30);

  ◦ *Renal:* Chronic kidney disease stage ≥3;

  ◦ *Infections:* Central nervous system/pneumonia/COVID-19/urinary.

## Model development and validation

*A priori* clinical expertise and data-driven variable selection will refine the variable list for modelling [4]. A blinded variable selection strategy [4] will be implemented *a priori* to reduce the list of CPM predictors down to a clinically pragmatic list of seven variables (or less) [19,20]. The primary driver of which variables are selected will be Delphi-method-driven consensus amongst the clinical research team and our Public Advisors (people affected by epilepsy) on which predictors are most important clinically and are readily accessible in the datasets (i.e., missing data patterns will be incorporated into the decision-making) [4,38]. Consensus will be reached through a series of workshops (i.e., focus groups) between the clinical research team and Public Advisors. A Delphi consensus exercise has the potential to highlight predictors beyond the scope of existing literature [38]. The focus will not be on the strength of the predictor (e.g., using backwards selection). This is because it is better to select predictors based on a wider body of clinical knowledge than to try to depend on statistical significance of results which may be sensitive to random variation in the data points due to sampling variability, as detailed elsewhere [39]. We will consider penalisation methods (e.g., *lasso*) [40,41], where appropriate.

CPRD will be used for model derivation. External validation will be undertaken in SAIL, CIPHA and TriNetX, maximising our CPM's international generalisability [16]. Multiple imputation will account for missing data [4].

Modelling will be done in three ways [17,27]:

(i) consider each binary outcome (A = seizure-related ED or hospital admission, and B = epilepsy-related death) separately, and develop individual prediction models for each using logistic regression [4,17];

(ii) define a composite outcome of A and B and develop a single logistic regression model for this [4,17];

(iii) model the outcomes sequentially through time [27]. For this, we let each outcome (A or B) be a state within a multi-state model [27] where individuals start in an initial (PWE) state, and we then model their risk of moving to A or B states (respecting the temporal order of these). Such an approach also allows us to model the competing risk of death from other causes (co-extracted from death records) and the competing risk of ED or hospital admission unrelated to seizures (also available). Approach (iii) will also allow us to model the interplay between admission and death and is thus our preferred approach [27], but we consider (i) and (ii) as computationally easier alternatives.

Internal validation within CPRD will involve 1,000 bootstrap analysis samples. Nagelkerke's R2 and Brier Scores will estimate overall model performance. Calibration will be estimated with calibration intercept, slope, and plots. Discrimination will be estimated with area under curve. Calibration and discrimination will be reassessed within SAIL, CIPHA and TriNetX to externally validate model performance [16].

A sample size calculation is shown in Table 1 [33].

## Model dissemination

As per our guide [18], we will explore various presentation formats for the externally validated CPM including points-based systems, graphical score charts, and nomograms. This aspect will be co-developed with members of the public through Patient and Public Involvement (PPI) forums organised by NIHR Applied Research Collaboration North West Coast (ARC NWC), and through consultation with potential end-users including clinical and research colleagues and EHR vendors by presenting the project at research conferences [18]. A final version of the CPM will be disseminated open-access in peer-reviewed journals and online through, e.g., www.mdcalc.com and GitHub (see, e.g., predictepilepsy.github.io and https://seds-tool.github.io/seds). These channels will facilitate external reviews and local adaptations of the CPM to be developed by external parties globally and also facilitate automated primary care system integration in future projects.

**Table 1. Sample size calculation.**

| Method | We performed a sample size calculation for each outcome based on our previous methodology [33]. We used prevalence figures for each outcome from prior literature [3,12], supported by our pre-submission CPRD feasibility search. |
|---|---|
| Logistic regression modelling assumptions | |
| (i) | The models have a predictive performance of either Nagelkerke's R2 of 15% or achieve at least a c-statistic of 0.6 (whichever leads to higher sample size to give a conservative estimate). |
| (ii) | We target a maximum degree of shrinkage/overfitting of 0.9. |
| (iii) | We consider a maximum of 50 candidate predictors per model. |
| Outcomes based these assumptions | |
| A | Based on a seizure-related ED or hospital admission prevalence of 8.8% [3], developing a logistic regression model for this outcome will require a minimum sample size of 44,750 if the model achieves at least a c-statistic of 0.6, or 6,430 if the model achieves a Nagelkerke's R2 of 15%. |
| B | Based on an epilepsy-related deaths prevalence of 15.4% [3,12], developing a logistic regression model for this outcome will require a minimum sample size of 27,528 if the model achieves at least a c-statistic of 0.6, or 4,951 if the model achieves a Nagelkerke's R2 of 15%. |
| Feasibility | |
| | We have already undertaken a feasibility search demonstrating a sample size of 227,271 PWE aged ≥16 years within CPRD between 01/01/2010–31/12/2021. Therefore, CPRD will be of more than sufficient size to develop the models for outcome definitions (i) and (ii). At present there are no sample size criteria for developing multi-state prediction models (for outcome definition (iii)) [27]. However, given the size of CPRD, we do not expect this to be a concern. SAIL holds an annual number of approximately 25,000 PWE [13]. Similar numbers are expected in CIPHA. We have identified 271,172 PWE in TriNetX [36]. These numbers will be sufficient for external validation in each dataset [33]. |

## Patient and public involvement and engagement (PPIE)

Our authorship team includes two Public Advisors from ARC NWC, who are members of the public affected by epilepsy. They have helped co-develop the study protocol and will co-produce the study with us. This will include helping to select appropriate prediction tool variables and presentation strategies, co-authoring journal manuscripts, co-leading public engagement workshops, and attending monthly project steering group meetings. Our regular PPIE workshops will also allow others affected by epilepsy to feedback on study design and implementation. We will engage wider public through social media. We have presented the study protocol at an ARC NWC PPIE forum, observing consensus among members of the public on need for the study.

### Ethics and dissemination

This project was deemed not to require ethics approval by The University of Liverpool's Central University Research Committee D as it does not involve human participants, human tissue or personal data (Reference 13789 22/03/2024). A final version of the CPM will be disseminated open-access in peer-reviewed journals and made freely available online, as described in more detail earlier.

### Data management plan

Data will be curated through University of Liverpool's (UoL) Active Data Storage -a centralised, secure, supported data storage facility with multiple layers of protection. Data are replicated between two secure physical locations and backed up regularly. A regular tape backup is made to a third physical location, and segregated from the public network both physically and logically. Data are encrypted in transit using SSL.

We will use a public repository (www.github.com) to make all diagnostic and outcome coding algorithms, metadata, and R analysis scripts used publicly available, facilitating external replication and adaptation.

All data storage and use will comply with legal obligations (including GDPR) and UoL's Research Data Management Policy.

## Discussion

This large study will develop and validate a CPM for people with epilepsy, creating an internationally generalisable tool for subsequent clinical implementation. The model predicts two important and interrelated outcomes of seizure-related emergency department or hospital admission, and epilepsy-related death. The latter is highlighted as a key recommendation for epilepsy research by NICE [26]. The study has been co-developed by epilepsy researchers and members of the public. Whilst the study protocol is limited to developing and externally validating a stand-alone CPM only, this will be sufficient to allow external users to adopt and implement the CPM into their local clinical care settings, as highlighted in the Prognosis Research Strategy (PROGRESS) 3 guidelines [16].

### Future directions

Further work will be required to evaluate how the model performs when implemented in real-world clinical settings, including its influence on decision-making and patient outcomes [16]. Whilst externally validating our CPM in geographically distinct Welsh SAIL and international TriNetX datasets will facilitate its clinical implementation sufficiently for external users [16,19,20], the use of CIPHA will enhance the potential impact of that clinical implementation further. This is because CIPHA would provide a means for us to feed our externally validated CPM directly back into clinical workflows electronically within Cheshire and Merseyside, allowing us to understand the impact of CPMs like this in future [42–44]. Through CIPHA, in future projects, we would be able to implement our CPM for population stratification via electronic dashboards with re-identification of patients directly to attending clinicians, providing them with targeted notifications about risk [42,43]. Furthermore, using CIPHA would allow us to track model performance over time and make evidence-based updates as new data become available. This approach would improve the model's reliability in real-world settings, particularly during periods of increased healthcare demand, such as winter [45]. To avoid overlap, we would remove Cheshire and Merseyside (~25,000 people with epilepsy (PWE)) from our CPRD development dataset (227,271 PWE) – unaffecting of our sample size calculations.

### Study status and timeline

This study remains at protocol submission stage. We plan to fund the importing of study data and commence data analysis within 12 months of protocol publication.

## Acknowledgments

We thank our Public Advisors for support with this study, and NIHR Applied Research Collaboration North West Coast for providing access to Public Advisors. We are also grateful to Tonina Takova for support with graphic design.

## Author contributions

**Conceptualization:** Gashirai K. Mbizvo, Glen P. Martin, Iain Buchan, Anthony G. Marson.

**Investigation:** Gashirai K. Mbizvo.

**Methodology:** Gashirai K. Mbizvo, Glen P. Martin, Laura J. Bonnett, Pieta Schofield, W Owen Pickrell, Iain Buchan, Gregory Y.H. Lip, Anthony G. Marson.

**Project administration:** Pieta Schofield.

**Resources:** W Owen Pickrell, Gregory Y.H. Lip, Anthony G. Marson.

**Writing – original draft:** Gashirai K. Mbizvo, Pieta Schofield.

**Writing – review & editing:** Gashirai K. Mbizvo, Glen P. Martin, Laura J. Bonnett, Hilary Garret, Alan Griffiths, W Owen Pickrell, Iain Buchan, Gregory Y.H. Lip, Anthony G. Marson.

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
