## [Decision Letter · Decision Letter 0]

30 Jul 2024

Dear Dr. Mbizvo,

Thank you for submitting your manuscript to PLOS ONE. After careful consideration, we feel that it has merit but does not fully meet PLOS ONE’s publication criteria as it currently stands. Therefore, we invite you to submit a revised version of the manuscript that addresses the points raised during the review process.

We look forward to receiving your revised manuscript.

Kind regards,

Muhammad Junaid Farrukh

Academic Editor

PLOS ONE

Reviewers' comments:

Reviewer's Responses to Questions

**Comments to the Author**

1. Does the manuscript provide a valid rationale for the proposed study, with clearly identified and justified research questions?

Reviewer #1: Yes

Reviewer #2: Yes

2. Is the protocol technically sound and planned in a manner that will lead to a meaningful outcome and allow testing the stated hypotheses?

Reviewer #1: Yes

Reviewer #2: Yes

3. Is the methodology feasible and described in sufficient detail to allow the work to be replicable?

Reviewer #1: Yes

Reviewer #2: Yes

4. Have the authors described where all data underlying the findings will be made available when the study is complete?

Reviewer #1: Yes

Reviewer #2: Yes

5. Is the manuscript presented in an intelligible fashion and written in standard English?

Reviewer #1: Yes

Reviewer #2: No

You may also provide optional suggestions and comments to authors that they might find helpful in planning their study.

Reviewer #1: The title is somewhat lengthy and complex, which might make it less accessible to a broader audience. Simplifying the language or shortening the title could make it more digestible.

The keywords you've provided for the study are relevant, however, increase specificity, avoid redundancy and decreases the numbers up to 5 only.

The abstract is quite lengthy for a standard abstract, which typically aims to succinctly communicate the study's essential elements. It delves into specific details, such as the number of patients in different databases, which might be more appropriate for the main text of the paper or the methods section.

While the introduction is divided into sections, each section is heavily loaded with information, some of which delve into specific details that may distract from the primary focus of the introduction. Each section is clearly labeled (e.g., Clinical prediction models, Seizure-related emergency department or hospital admissions, Epilepsy-related deaths), which is helpful. However, the narrative could more explicitly connect these sections to show how they build on each other to justify the study's necessity and aims more coherently.

Ethics and dissemination AND Data management plan should be in the method section

please re write the discussion section

Reviewer #2: You need to improve English used. Moreover, you need to write more information about your work

**Do you want your identity to be public for this peer review?** For information about this choice, including consent withdrawal, please see our Privacy Policy

Reviewer #1: No

Reviewer #2: **Yes: ** Noor Kifah Al-Tameemi

---

## [Author Response · Author response to Decision Letter 1]

7 Aug 2024

Response to Reviewers

Author response: We have now formatted the manuscript to the journal’s specifications.

Author response: Our ethics statement is in the specified correct location in the manuscript.

Reviewer comments:

5. Is the manuscript presented in an intelligible fashion and written in standard English?

Reviewer #1: Yes

Reviewer #2: No

Author response: We thank the reviewers for these responses. We would like to respectfully point out that according to the journal's guidelines as stated above, reviewers should specify any typographical or grammatical errors if they select "no" to this question. Reviewer #2 has not noted any specific errors, so we are unable to address them in this revision. Reviewer #1 feels the manuscript presented in an intelligible fashion and written in standard English, which is reassuring.

Reviewer #1: The title is somewhat lengthy and complex, which might make it less accessible to a broader audience. Simplifying the language or shortening the title could make it more digestible.

Author response: This is helpful suggestion, thank you. We have now shortened and simplified the title to the following: “Developing and validating a clinical prediction model to predict epilepsy-related hospital admission or death: a cohort study protocol”

Reviewer #1: The keywords you've provided for the study are relevant, however, increase specificity, avoid redundancy and decreases the numbers up to 5 only.

Author response: Noted with thanks. We have now amended the keywords to the following five: “prognostic model, epilepsy health data, accident and emergency attendance, inpatient admission, mortality”

Reviewer #1: The abstract is quite lengthy for a standard abstract, which typically aims to succinctly communicate the study's essential elements. It delves into specific details, such as the number of patients in different databases, which might be more appropriate for the main text of the paper or the methods section.

Author response: These points are fair and well received. We have now removed the following from the abstract: “Data are held for 60 million patients in England on CPRD, 3.1m in Wales on SAIL, 2.6m in Cheshire and Merseyside on CIPHA, and 250m across 19 countries in TriNetX” - and replaced it with “Sample sizes of over 100,000 PWE are expected across these research platforms” – which is much simpler.

Reviewer #1: While the introduction is divided into sections, each section is heavily loaded with information, some of which delve into specific details that may distract from the primary focus of the introduction. Each section is clearly labeled (e.g., Clinical prediction models, Seizure-related emergency department or hospital admissions, Epilepsy-related deaths), which is helpful. However, the narrative could more explicitly connect these sections to show how they build on each other to justify the study's necessity and aims more coherently.

Author response: We have now re-ordered and re-written the whole introduction, which now flows much more seamlessly and reads as follows:

Introduction

Hospital admissions and deaths from seizures are important outcomes to people with epilepsy (PWE). To inform patient counselling and clinical guidance about such outcomes, statistically robust and clinically meaningful clinical prediction models (CPMs) of who is at high (and low) risk of these outcomes can be developed and validated.

Seizure-related emergency department or hospital admissions

As we previously demonstrated [1], PWE are at substantially increased risk of seizure-related emergency department (ED) or hospital admission [2]. Indeed, seizures are the most common neurological cause of ED or hospital admission in England [2], and such admissions can be predictors of subsequent epilepsy-related death [3, 4]. However, undergoing ED or hospital admission for epilepsy is often clinically unnecessary and typically leads to little benefit for management of the epilepsy because treatment decisions in epilepsy are complex and require specialist expertise, training and guidance [2]. The admitted PWE are usually seen by junior doctors and physicians without particular expertise in epilepsy and frequently discharged without specialist consultation or referral [1, 2, 5], representing missed opportunity for risk mitigation at a time when this may be crucial [1]. Such admissions could be avoided by early prediction of high-risk groups (through providing non-specialists such as general practitioners (GPs), paramedics, and physicians with tools for early prediction) because epilepsy is an ambulatory-care-sensitive condition [2, 6, 7]; meaning the high-risk groups are amenable to preventive targeting with scheduled specialist and community resources, diversion to alternative care pathways, optimised self-management, and accelerated medication reviews [8, 9].

Epilepsy-related deaths

As we previously demonstrated [5, 10], PWE are at significantly increased risk of premature death. Some of those deaths may be entirely unrelated to their epilepsy. However, a substantial proportion are epilepsy-related [10, 11]. These are operationally defined as any death listing epilepsy as an underlying or contributory cause within death records [5, 12, 13], in line with national guidance [14]. Although sudden unexpected death in epilepsy (SUDEP) is a common epilepsy-related death, we have shown it is equally common for epilepsy-related death to occur through other mechanisms including aspiration pneumonia, cardiac arrest, antiseizure medication (ASM) poisoning, drowning, and alcohol dependence [5, 10]. Most epilepsy-related deaths occur in young adults [5, 10]. When we looked at epilepsy-related deaths as a group, nearly 80% were potentially avoidable [5]. Therefore, it is becoming increasingly pragmatic to investigate epilepsy-related deaths as a group rather than focusing on each cause individually [11, 12, 15].

Clinical prediction models (CPMs)

CPMs estimate the risk of a future clinical outcome in an individual patient based on information retrieved from their routinely collected health data including age, sex, ethnicity, comorbidities, previous treatments, and other characteristics [16]. Therefore, CPMs help guide and standardise shared clinical decision-making and service delivery by drawing attention to high-risk individuals (needing earlier or more invasive treatments), and low-risk individuals (needing less invasive treatments with fewer side-effects) [17]. This helps personalise clinical management strategies. CPMs also help patients understand their own risks by quantifying them in a succinctly presented tool to aid patient-doctor discussions about risk [18]. For example, our CHA₂DS₂-VASc score [19, 20] is a simple 7-item CPM using everyday clinical information to accurately predict risk of stroke after developing atrial fibrillation (AF). This helps guide subsequent anticoagulation treatment decisions and patient discussions. We made CHA₂DS₂-VASc freely available online to any clinician in the world at www.mdcalc.com. This combination of simplicity of use, accuracy, and wide availability has translated into substantial global impact and recommended use of this CPM within National Institute for Health and Care Excellence (NICE) guidelines [21].

CPMs for clinically meaningful outcomes for PWE

An effective way to help avoid epilepsy-related deaths is to develop CPMs to identify high-risk groups and thereby aid clinicians in prioritising their care [4]. For example, in non-specialist settings, where such patients often first present [1, 2], high CPM scores could prompt re-discussion about seizure safety, signposting to online resources for support (e.g. www.epilepsy.org.uk/living/safety), and rapid-access epilepsy clinic referral (rather than reserving such clinics for first-fit patients alone). The potential impact of such approaches is well recognised. For example, the asthma deaths review demonstrated that many deaths were likely to be preventable simply through better proactive clinical management, such as arranging for patients to be seen promptly after a hospital admission, and through following clinical guidelines [22]. In specialist settings, high CPM scores could prompt referral for epilepsy surgery earlier than would have otherwise been considered, new discussion in a multidisciplinary team meeting, further medication reviews, or implementation of seizure alarms and nocturnal supervision.

There are currently no CPMs for ED or hospital admissions in PWE. Work is planned to develop a risk prediction tool to estimate the benefits a person with epilepsy would receive if conveyed to ED and risks if not [8]. Additionally, there are few CPMs of mortality in PWE. Although our recently developed Scottish Epilepsy Deaths Study score [4] predicted epilepsy-related deaths as a group, its reliance on hand-searched medical records limited sample sizes to 224 cases/controls, which widened confidence intervals, reduced precision, and meant only four predictors could be included. The SUDEP-7 and SUDEP-3 inventories focused on predicting SUDEP alone [23], and therefore missed the remaining epilepsy-related deaths as a group. They were also drawn from small sample sizes of 19 and 28 patients, respectively, limiting their reliability. Similarly, the Personalised Prediction Tool [24] focused on SUDEP alone and had only 287 cases and 986 controls.

None of the above CPMs have been validated in independent, geographically distinct, datasets (so-called external validation), meaning they cannot currently be used in clinical practice [16]. The SUDEP and Seizure Safety Checklist [25] is expert consensus guidance rather than a CPM. Developing a validated CPM for epilepsy-related deaths was highlighted as a key recommendation for epilepsy research in recent NICE guidelines [26].

Study aims and potential benefits to PWE

In line with the recommendations of NICE [26], we aim to develop and externally validate a CPM to predict the joint risk of two outcomes occurring within the next year in PWE [27]: A) seizure-related ED or hospital admission; and B) epilepsy-related death. We hypothesise the CPM will be able to identify individuals at high or low risk of either or both outcomes. This will give clinicians seeing PWE a validated tool to predict either or both of these important outcomes. This has not been done before despite both outcomes being common, potentially avoidable, devastating for patients and their families, and closely interrelated [1-5, 9]. We will develop guidance for clinicians on proposed actions to take based on the overall risk score. When implemented, we expect this work to contribute to a reduction in the number of seizure-related ED or hospital admissions and epilepsy-related deaths.

Reviewer #1: Ethics and dissemination AND Data management plan should be in the method section.

Author response: Thanks. Ethics and dissemination was already located in the methods. We have now moved the data management plan to the methods section as well.

Reviewer #1: please re write the discussion section.

Author response: We appreciate the reviewer's feedback. To effectively revise the discussion section, we would need specific guidance on the issues that need to be addressed.

Reviewer #2: You need to improve English used. Moreover, you need to write more information about your work

Author response: We appreciate the reviewer's feedback. To improve the English, we need specific instances where the language is inadequate. Additionally, we require more detailed guidance on the additional information needed about our work to ensure we address the reviewer's concerns accurately.

---

## [Decision Letter · Decision Letter 1]

8 Oct 2024

Dear Dr. Mbizvo,

Thank you for submitting your manuscript to PLOS ONE. After careful consideration, we feel that it has merit but does not fully meet PLOS ONE’s publication criteria as it currently stands. Therefore, we invite you to submit a revised version of the manuscript that addresses the points raised during the review process.

We look forward to receiving your revised manuscript.

Kind regards,

Francesco Deleo, MD

Academic Editor

PLOS ONE

Reviewers' comments:

Reviewer's Responses to Questions

**Comments to the Author**

1. Does the manuscript provide a valid rationale for the proposed study, with clearly identified and justified research questions?

Reviewer #1: Yes

Reviewer #2: Yes

2. Is the protocol technically sound and planned in a manner that will lead to a meaningful outcome and allow testing the stated hypotheses?

Reviewer #1: Yes

Reviewer #2: Yes

3. Is the methodology feasible and described in sufficient detail to allow the work to be replicable?

Reviewer #1: Yes

Reviewer #2: Yes

4. Have the authors described where all data underlying the findings will be made available when the study is complete?

Reviewer #1: Yes

Reviewer #2: Yes

5. Is the manuscript presented in an intelligible fashion and written in standard English?

Reviewer #1: Yes

Reviewer #2: Yes

You may also provide optional suggestions and comments to authors that they might find helpful in planning their study.

Reviewer #1: 1-Expanding on the rationale behind selecting these specific datasets (CPRD, SAIL, CIPHA, TriNetX) and how they cover a representative population would enhance the justification of the study's scope.

2- More explicit guidelines on how predictors will be chosen and defined would ensure clarity and reproducibility.

3-Some sentences are overly long and complex. Breaking these down into simpler sentences would improve readability, especially for non-expert readers.

4-Ensure consistency in terminology (e.g., "epilepsy-related death" vs. "epilepsy-related mortality") to avoid confusion.

5-There are minor grammatical issues (e.g., subject-verb agreement, redundant phrases) that should be corrected in revision.

Reviewer #2: ….

**Do you want your identity to be public for this peer review?** For information about this choice, including consent withdrawal, please see our Privacy Policy

Reviewer #1: No

Reviewer #2: **Yes: ** Noor Kifah Al-Tameemi

---

## [Author Response · Author response to Decision Letter 2]

12 Nov 2024

REVIEWER 1.1

Reviewer #1: 1-Expanding on the rationale behind selecting these specific datasets (CPRD, SAIL, CIPHA, TriNetX) and how they cover a representative population would enhance the justification of the study's scope.

AUTHOR 1.1

Thank you for this helpful suggestion. We have now included the following justification, on page 6:

“There are several reasons for planning to study CPRD, SAIL, CIPHA and TriNetX. First, it provides a contingency plan should we fail acquire one or more of the datasets, e.g. owing to cost. Second, it allows for internal and external validation within and between the geographically distinct populations represented by each of these datasets, allowing the final model generated to be more widely generalisable, facilitating future clinical implementation. In particular, each dataset’s coverage adds value to understanding the model’s performance across different healthcare systems and demographics. For example, CPRD is strengthened by its extensive primary care representation, SAIL by its linkage to socio-demographic data, CIPHA allows real-time analytics with re-identification at source, and TriNetX has a substantial sample size, offering flexibility in selecting a large number of predictors.”

REVIEWER 1.2

2- More explicit guidelines on how predictors will be chosen and defined would ensure clarity and reproducibility.

AUTHOR 1.2

Thanks. We have now expanded the model development and validation section on page 8 as follows:

“A priori clinical expertise and data-driven variable selection will refine the variable list for modelling [4]. A blinded variable selection strategy [4] will be implemented a priori to reduce the list of CPM predictors down to a clinically pragmatic list of seven variables (or less) [19, 20]. The primary driver of which variables are selected will be Delphi-method-driven consensus amongst the clinical research team and our Public Advisors (people affected by epilepsy) on which predictors are most important clinically and are readily accessible in the datasets (i.e. missing data patterns will be incorporated into the decision-making) [4, 37]. Consensus will be reached through a series of workshops between the clinical research team and Public Advisors. A Delphi consensus exercise has the potential to highlight predictors beyond the scope of existing literature [37]. The focus will not be on the strength of the predictor (e.g. using backwards selection). This is because it is better to select predictors based on a wider body of clinical knowledge than to try to depend on statistical significance of results which may be sensitive to random variation in the data points due to sampling variability, as detailed elsewhere [38]. We will consider penalisation methods (e.g., lasso), [39, 40] where appropriate.”

REVIEWER 1.3

3-Some sentences are overly long and complex. Breaking these down into simpler sentences would improve readability, especially for non-expert readers.

AUTHOR 1.3

We have now simplified the relevant sentences as follows:

Page 2:

Old version

This retrospective open cohort study develops and externally validates a clinical prediction model (CPM) to predict the joint risk of two important outcomes occurring within the next year in people with epilepsy (PWE): A) seizure-related emergency department or hospital admission; and B) epilepsy-related death.

New version

This retrospective open cohort study develops and externally validates a clinical prediction model (CPM) to predict the joint risk of two important outcomes occurring within the next year in people with epilepsy (PWE). These are: A) seizure-related emergency department or hospital admission; and B) epilepsy-related death.

Page 2:

Old version

Routinely collected, anonymised, electronic health data from Clinical Practice Research Datalink (CPRD), Secure Anonymised Information Linkage databank (SAIL), Combined Intelligence for Population Health Action (CIPHA), and TriNetX research platforms will be used to study PWE aged ≥16 years having outcomes A and/or B between 2010–2022.

New version

Routinely collected, anonymised, electronic health data from Clinical Practice Research Datalink (CPRD), Secure Anonymised Information Linkage databank (SAIL), Combined Intelligence for Population Health Action (CIPHA), and TriNetX research platforms will be used. We will study PWE aged ≥16 years having outcomes A and/or B between 2010–2024 within these datasets.

Page 8

Old version

The focus will not be on the strength of the predictor (e.g. using backwards selection) as it is better to select predictors based on a wider body of clinical knowledge than to try to depend on statistical significance of results which may be sensitive to random variation in the data points due to sampling variability, as detailed elsewhere [37].

New version

The focus will not be on the strength of the predictor (e.g. using backwards selection). This is because it is better to select predictors based on a wider body of clinical knowledge than to try to depend on statistical significance of results which may be sensitive to random variation in the data points due to sampling variability, as detailed elsewhere [38].

REVIEWER 1.4

4-Ensure consistency in terminology (e.g., "epilepsy-related death" vs. "epilepsy-related mortality") to avoid confusion.

AUTHOR 1.4

Thanks. We have changed to now consistently report epilepsy-related death rather than epilepsy-related mortality, throughout the text.

REVIEWER 1.5

5-There are minor grammatical issues (e.g., subject-verb agreement, redundant phrases) that should be corrected in revision.

AUTHOR 1.5

We have corrected minor grammatical issues which we have been able identify. For example, on page 11:

Old version

Whilst the study protocol is limited to developing and externally validating a stand-alone CPM only, this will be sufficient to allow external users to consider adopting and implementing the CPM into their local clinical care settings, as highlighted in the Prognosis Research Strategy (PROGRESS) 3 guidelines [16].

New version

Whilst the study protocol is limited to developing and externally validating a stand-alone CPM only, this will be sufficient to allow external users to adopt and implement the CPM into their local clinical care settings, as highlighted in the Prognosis Research Strategy (PROGRESS) 3 guidelines [16].

---

## [Decision Letter · Decision Letter 2]

28 Jan 2025

Dear Dr. Mbizvo,

Thank you for submitting your manuscript to PLOS ONE. After careful consideration, we feel that it has merit but does not fully meet PLOS ONE’s publication criteria as it currently stands. Therefore, we invite you to submit a revised version of the manuscript that addresses the points raised during the review process.

https://journals.plos.org/plosone/s/submission-guidelines#loc-laboratory-protocols . Additionally, PLOS ONE offers an option for publishing peer-reviewed Lab Protocol articles, which describe protocols hosted on protocols.io. Read more information on sharing protocols at https://plos.org/protocols?utm_medium=editorial-email&utm_source=authorletters&utm_campaign=protocols .

We look forward to receiving your revised manuscript.

Kind regards,

Francesco Deleo, MD

Academic Editor

PLOS ONE

**Journal Requirements:**

**Additional Editor Comments:**

The reviewers found the manuscript text improved compared to the first submission. However, there are still some requests for clarification on the protocol and several suggestions for an English revision

Reviewers' comments:

Reviewer's Responses to Questions

**Comments to the Author**

1. Does the manuscript provide a valid rationale for the proposed study, with clearly identified and justified research questions?

Reviewer #2: Yes

Reviewer #3: Partly

Reviewer #4: Yes

2. Is the protocol technically sound and planned in a manner that will lead to a meaningful outcome and allow testing the stated hypotheses?

Reviewer #2: Yes

Reviewer #3: Partly

Reviewer #4: Yes

3. Is the methodology feasible and described in sufficient detail to allow the work to be replicable?

Reviewer #2: Yes

Reviewer #3: Yes

Reviewer #4: Yes

4. Have the authors described where all data underlying the findings will be made available when the study is complete?

Reviewer #2: Yes

Reviewer #3: Yes

Reviewer #4: Yes

5. Is the manuscript presented in an intelligible fashion and written in standard English?

Reviewer #2: Yes

Reviewer #3: No

Reviewer #4: Yes

You may also provide optional suggestions and comments to authors that they might find helpful in planning their study.

**Reviewer #2: ** I don’t have any comment to authors

**Reviewer #3:**  In this manuscript the authors developed and evaluated a clinical prediction model (CPM) useful to predict epilepsy-related hospital admission and epilepsy-related mortality. The topic of the present study could be of interest to the reader, since research on this topic could increase the knowledge about the tool to manage fatal outcomes in epilepsy sector. However, this study presents some limitations: i) Some sentences are too long and contain technical information; ii) All the abbreviations should be reported extensively the first time they are mentioned (e.g ‘AF’ for ‘Atrial Fibrillation or “COPD” for “Chronic obstructive pulmonary disease”); iii) Check in the text (including “abstract”) the terms “epilepsy-related death” and “epilepsy-related mortality” trying to make the whole text homogeneous; iv) The Delphi methods steps should be clarify (please define what do you mean with “a series of workshops”, maybe “focus groups” or “group of discussion” could be more appropriate).” v) Discussions needs to be revised in order to be more informative and less vague/superficial. Finally, the paper is not easy to read and a revision about the English language as well as the technical terms is necessary.

**Reviewer #4: ** The present study protocol aims to develop and validate a clinical prediction model to predict epilepsy-related hospital admission or death.

The revised version of the manuscript is clearly-written and the methods are well-detailed. I found this protocol very interesting to read and I think that are a lot of implications for clinical practice and research.

There are a few minor points/comments for the authors:

-Title: As the so-called outcome A includes ‘seizure-related emergency department’ together with ‘hospital admission’, I would suggest to include also the first in the title.

-Abstract-Methods: ‘Candidate predictors will include demographic, lifestyle, clinical, and management’. I would add ‘variables’ at the end of the statement.

-Abstract- Conclusions: I would suggest not to the start the section with ‘This is the largest study…’. Please, revise this and other similar statements along the manuscript.

-Lay summary- first row: ‘Some people…’ I would include the exact number, if possible. ‘Some’ is vague.

Fourth row: Again, please revise the statement: ‘Our study is the first to do this’.

Row 11: ‘Giving clinicians the tool […]’. I would replace the latter with ‘Providing clinicians with the tool […]’.

-Introduction-Row 1: ‘[…] outcomes to people […]’. I would replace the latter with ‘[…] for people […]’.

‘Seizure-related emergency department or hospital admissions’: check for consistency of using the plural ‘admissions’ along the manuscript.

‘CPMs for clinically meaningful outcomes for PWE’- row 6: ‘For example, the asthma deaths review […]’. I would say ‘[…] an asthma death review […]’.

-Methods-study design-first 10 rows: I would reshape these by listing datasets using bullet points for each dataset.

Importantly, I would suggest to use the TRIPOD checklist and include in a relevant section of the protocol.

-Discussion- row 1: ‘This is the largest study […]: please, revise this (see my comment above).

**Do you want your identity to be public for this peer review?** For information about this choice, including consent withdrawal, please see our Privacy Policy

Reviewer #2: No

Reviewer #3: No

Reviewer #4: **Yes: ** Andrea Giordano

---

## [Author Response · Author response to Decision Letter 3]

24 Jun 2025

Reviewer #2: I don’t have any comment to authors

Reviewer #3: In this manuscript the authors developed and evaluated a clinical prediction model (CPM) useful to predict epilepsy-related hospital admission and epilepsy-related mortality. The topic of the present study could be of interest to the reader, since research on this topic could increase the knowledge about the tool to manage fatal outcomes in epilepsy sector. However, this study presents some limitations:

Reviewer 3.1:

Some sentences are too long and contain technical information

Author 3.1:

Thank you. We have carefully reviewed the manuscript and shortened or restructured all overly long sentences, often by splitting them into shorter, clearer components. We have also simplified technical phrasing where appropriate. These edits are tracked in the revised manuscript.

Reviewer 3.2:

All the abbreviations should be reported extensively the first time they are mentioned (e.g ‘AF’ for ‘Atrial Fibrillation or “COPD” for “Chronic obstructive pulmonary disease”)

Author 3.2:

We have reviewed the manuscript and ensured that all abbreviations are spelled out in full on first use. The revised text reflects these changes.

Reviewer 3.3:

Check in the text (including “abstract”) the terms “epilepsy-related death” and “epilepsy-related mortality” trying to make the whole text homogeneous;

Author 3.3:

Thank you. We have carefully reviewed the manuscript, including the abstract, and confirm that we consistently use the term “epilepsy-related death” throughout. The term “epilepsy-related mortality” does not appear in the text.

Reviewer 3.4:

The Delphi methods steps should be clarify (please define what do you mean with “a series of workshops”, maybe “focus groups” or “group of discussion” could be more appropriate).”

Author 3.4:

We have revised the text to clarify that the “series of workshops” refers to structured focus groups involving clinicians and public advisors, in line with Delphi methodology.

Reviewer 3.5:

Discussions needs to be revised in order to be more informative and less vague/superficial.

Author 3.5:

Thank you. We have reviewed the Discussion section and made minor changes to improve clarity and specificity, particularly regarding how the model may be implemented and updated over time. We hope this addresses the reviewer’s concern.

Reviewer 3.6:

Finally, the paper is not easy to read and a revision about the English language as well as the technical terms is necessary.

Author 3.6:

We have made substantial revisions to improve readability, including simplifying technical language and restructuring long sentences. We hope the manuscript now reads more clearly and is easier to follow.

Reviewer #4: The present study protocol aims to develop and validate a clinical prediction model to predict epilepsy-related hospital admission or death.

The revised version of the manuscript is clearly-written and the methods are well-detailed. I found this protocol very interesting to read and I think that are a lot of implications for clinical practice and research.

There are a few minor points/comments for the authors:

Reviewer 4.1

-Title: As the so-called outcome A includes ‘seizure-related emergency department’ together with ‘hospital admission’, I would suggest to include also the first in the title.

Author 4.1

Many thanks. Both terms have been added to the title.

Reviewer 4.2

-Abstract-Methods: ‘Candidate predictors will include demographic, lifestyle, clinical, and management’. I would add ‘variables’ at the end of the statement.

Author 4.2

Thanks. We have added ‘variables’ at the end of the statement.

Reviewer 4.3

-Abstract- Conclusions: I would suggest not to the start the section with ‘This is the largest study…’. Please, revise this and other similar statements along the manuscript.

Author 4.3

Thanks. We have revised these accordingly to “large study” and removed the superlatives.

Reviewer 4.4

-Lay summary- first row: ‘Some people…’ I would include the exact number, if possible. ‘Some’ is vague.

Author 4.4

Thank you. We agree that precise figures can be helpful; however, in this case, estimates vary depending on the population and data source. To avoid presenting potentially misleading figures, we have chosen to use the more general phrasing “people with epilepsy,” which accurately reflects the elevated risk without oversimplifying the data.

Reviewer 4.5

Fourth row: Again, please revise the statement: ‘Our study is the first to do this’.

Author 4.5

Thanks. We have removed this superlative.

Reviewer 4.6

Row 11: ‘Giving clinicians the tool […]’. I would replace the latter with ‘Providing clinicians with the tool […]’.

Author 4.6

Thanks. We have replaced the latter with ‘Providing clinicians with the tool’, as helpfully suggested.

Reviewer 4.7

-Introduction-Row 1: ‘[…] outcomes to people […]’. I would replace the latter with ‘[…] for people […]’.

Author 4.7

Thanks. We have replaced ‘to people’ with ‘for people’

Reviewer 4.8

‘Seizure-related emergency department or hospital admissions’: check for consistency of using the plural ‘admissions’ along the manuscript.

Author 4.8

Thanks. We have, for consistency, replaced these with the singular ‘admission’, where relevant, instead of ‘admissions’.

Reviewer 4.9

‘CPMs for clinically meaningful outcomes for PWE’- row 6: ‘For example, the asthma deaths review […]’. I would say ‘[…] an asthma death review […]’.

Author 4.9

Thanks. We have now used ‘an asthma death review’

Reviewer 4.10

-Methods-study design-first 10 rows: I would reshape these by listing datasets using bullet points for each dataset.

Author 4.10

Great idea. Thanks. We have now listed with bullet points.

Reviewer 4.11

Importantly, I would suggest to use the TRIPOD checklist and include in a relevant section of the protocol.

Author 4.11

We have used the checklist and added details to the methods.

Reviewer 4.12

-Discussion- row 1: ‘This is the largest study […]: please, revise this (see my comment above).

Author 4.12

Thanks. We have removed the superlatives.

---

## [Decision Letter · Decision Letter 3]

8 Jul 2025

Developing and validating a clinical prediction model to predict epilepsy-related emergency department attendance, hospital admission, or death: a cohort study protocol

PONE-D-24-11671R3

Dear Dr. Mbizvo,

We’re pleased to inform you that your manuscript has been judged scientifically suitable for publication and will be formally accepted for publication once it meets all outstanding technical requirements.

Kind regards,

Francesco Deleo, MD

Academic Editor

PLOS ONE

Additional Editor Comments (optional):

Reviewers' comments:

Reviewer's Responses to Questions

**Comments to the Author**

1. Does the manuscript provide a valid rationale for the proposed study, with clearly identified and justified research questions?

Reviewer #4: Yes

2. Is the protocol technically sound and planned in a manner that will lead to a meaningful outcome and allow testing the stated hypotheses?

Reviewer #4: Yes

3. Is the methodology feasible and described in sufficient detail to allow the work to be replicable?

Reviewer #4: Yes

4. Have the authors described where all data underlying the findings will be made available when the study is complete?

Reviewer #4: Yes

5. Is the manuscript presented in an intelligible fashion and written in standard English?

Reviewer #4: Yes

You may also provide optional suggestions and comments to authors that they might find helpful in planning their study.

Reviewer #4: I have carefully read the replies along with the changes in the main document and I have no further comments/issues.

**Do you want your identity to be public for this peer review?** For information about this choice, including consent withdrawal, please see our Privacy Policy

Reviewer #4: **Yes: ** Andrea Giordano

---

## [Editor Report · Acceptance letter]

PONE-D-24-11671R3

PLOS ONE

Dear Dr. Mbizvo,

I'm pleased to inform you that your manuscript has been deemed suitable for publication in PLOS ONE. Congratulations! Your manuscript is now being handed over to our production team.

Kind regards,

on behalf of

Dr. Francesco Deleo

Academic Editor

PLOS ONE